# A GFP Reporter MR766-Based Flow Cytometry Neutralization Test for Rapid Detection of Zika Virus-Neutralizing Antibodies in Serum Specimens

**DOI:** 10.3390/vaccines7030066

**Published:** 2019-07-16

**Authors:** Etienne Frumence, Wildriss Viranaicken, Gilles Gadea, Philippe Desprès

**Affiliations:** Université de La Réunion, INSERM U1187, CNRS UMR 9192, IRD UMR 249, Unité Mixte Processus Infectieux en Milieu Insulaire Tropical, Plateforme Technologique CYROI, 97491 Sainte-Clotilde, La Réunion, France

**Keywords:** arbovirus, Zika virus, GFP reporter ZIKV, flow cytometry, neutralization test, mouse immune sera, humoral immune response, virus neutralizing antibodies, plaque-reduction neutralization test, flow cytometry neutralization test

## Abstract

Zika virus (ZIKV) is an emerging arthropod-borne virus of major public health concern. ZIKV infection is responsible for congenital Zika disease and other neurological defects. Antibody-mediated virus neutralization is an essential component of protective antiviral immunity against ZIKV. In the present study, we assessed whether our GFP reporter ZIKV derived from African viral strain MR766 could be useful for the development of a flow cytometry neutralization test (FNT), as an alternative to the conventional plaque-reduction neutralization test (PRNT). To improve the efficacy of GFP-expressing MR766, we selected virus variant MR766^GFP^ showing a high level of GFP signal in infected cells. A MR766^GFP^-based FNT was assayed with immune sera from adult mice that received ZIKBeHMR-2. The chimeric ZIKV clone ZIKBeHMR-2 comprises the structural protein region of epidemic strain BeH819015 into MR766 backbone. We reported that adult mice inoculated with ZIKBeHMR-2 developed high levels of neutralizing anti-ZIKV antibodies. Comparative analysis between MR766^GFP^-based FNT and conventional PRNT was performed using mouse anti-ZIKBeHMR-2 immune sera. Indistinguishable neutralization patterns were observed when compared with PRNT_50_ and FNT_50_. We consider that the newly developed MR766^GFP^-based FNT is a valid format for measuring ZIKV-neutralizing antibodies in serum specimens.

## 1. Introduction

Mosquito-borne Zika virus (ZIKV) is an emerging flavivirus initially reported in Africa in 1947 [1]. Recently, ZIKV became a public health concern with recent epidemics occurring in Yap islands (2007), French Polynesia (2013), and South America (2015) [2]. During these outbreaks, ZIKV infection was reported to cause severe complications in humans, such as developmental abnormalities in infants and Guillain-Barre syndrome in adults [3]. Also, ZIKV is capable of long-term persistence in human body fluids and sexual transmission of disease has been documented [4,5]. 

Laboratory diagnosis of ZIKV infection is mainly based on RT-PCR method during the acute phase of the disease [5,6,7,8]. Serological diagnosis of ZIKV infection is obtained at least one week after the onset of the clinical symptoms [5,6]. It is frequently observed that members of flavivirus genus share strong antigenic cross-reactivity making inconclusive serological tests especially in regions where ZIKV and related flaviviruses, such as dengue viruses, co-circulate. To overcome this issue, the detection of antibodies that have neutralizing activity against ZIKV can be achieved to confirm recent infection [8]. Plaque-reduction neutralization test (PRNT) is a widely accepted method to measure neutralizing antibodies in serum specimens [9]. Although PRNT is considered as a “gold standard” assay for flavivirus neutralization, this specific but conventional format is a time-consuming method making difficult the screening of a large number of samples. Noteworthy, virus plaque-based analysis is limited to cultured cell lines, such as green monkey epithelial Vero cells, that permit plaque-forming assay. 

New biological tests revisiting the conventional PRNT method are necessary for the rapid measurement of ZIKV-neutralizing antibodies in large serum series. Consequently, several reporter gene-based methods for neutralizing assays have been recently developed for ZIKV [10,11,12]. Given that reporter systems afford the ability to visualize viral expression inside the infected cells, we generated a GFP reporter ZIKV entitled ZIKV_GFP_ [13]. We demonstrated that ZIKV_GFP_ is a performing tool for the monitoring of ZIKV replication in the host cell as well as the screening of antiviral compounds [13,14,15,16,17,18,19,20]. In the present study, we adapted ZIKV_GFP_ for the measurement of ZIKV-neutralizing antibodies by flow cytometry analysis. The GFP reporter ZIKV-based flow cytometry neutralization test (FNT) was challenged with serum samples from mice inoculated with ZIKV. Results showed that our newly developed FNT generated ZIKV-neutralizing antibody titers that are equivalent to those obtained with conventional PRNT.

## 2. Materials and Methods 

### 2.1. Cells and Virus

Vero cells (clone E6) and A549^Dual^ cells (Invivogen, Toulouse, France), referred to hereafter as “A549 cells”, were cultured at 37 °C under 5% CO_2_ atmosphere at 37 °C in MEM Eagle medium, supplemented with 5% heat-inactivated fetal bovine serum (FBS) and 2 mM _L_-glutamine, 1 mM sodium pyruvate, 100 U/mL penicillin, 0.1 mg/mL streptomycin, and 0.5 µg/mL amphotericin B (PAN Biotech, Aidenbach, Germany). The recombinant ZIKV_GFP_ derived from molecular clone ZIKV MR766^MC^ is described elsewhere [13]. ZIKV_GFP_ and derived viral clone MR766^GFP^ stocks were amplified on Vero cells and titrated by plaque-forming assay on Vero cells, as described elsewhere [21]. Virus titer was expressed as plaque-forming unit per mL (PFU/mL).

### 2.2. Generation of MR766^GFP^

Variants of ZIKV_GFP_ were selected using a limiting dilution cloning method. Briefly, Vero cells seeded in 96-well plates (10,000 cells/well) were infected 5 days with ZIKV_GFP_ at the multiplicity of infection (m.o.i) of l0^−4^. Supernatants of GFP-positive cell monolayers were recovered, and the second round of virus cloning was performed on Vero cells, as described above. Viral clone MR766^GFP^ was selected and then passaged twice on Vero cells to produce a final virus stock for experiments. The infectious titer of the working MR766^GFP^ stock was 7.7 log PFU/mL on Vero cells.

### 2.3. Focus-Forming Assay

For the focus-forming assay, Vero cells were seeded in 24-well plates. Virus samples were added to the cells for 2 h at 37 °C and then incubated with 0.8% carboxymethylcellulose (CMC). After a 4-day incubation, cell monolayers were fixed with 3.7% formaldehyde (FA) in PBS, followed by permeabilization with 0.15% Triton X-100 in PBS for 5 min. Cells were incubated with a mouse anti-GFP (Abcam, Cambridge, UK) at dilution 1:1000 followed by incubation with HRP-conjugated anti-mouse IgG antibody (Abcam, Cambridge, UK) at dilution 1:2000. Virus plaques were revealed with Vector NovaRED^TM^ peroxidase substrate (Cliniscences Nanterre, France) and counterstained with 0.5% crystal violet in 20% ethanol. Plates were scanned with an Epson scanner (Levallois-Perret, France).

### 2.4. Mouse Serum Specimens

The protocols and subsequent experiments in mice were ethically approved by the Ethics Committee for Control of Experiments on Animals at the CECEMA (Montpellier, France) with the reference n°036 and by the French Ministère de l’Enseignement Supérieur, de la Recherche et de l’Innovation with reference APAFIS#9137-2017030316134494 v6 (February 28th, 2018) [22,23]. A total of 29 individualized mouse immune serum specimens was selected for this study [23]. The first group of adult BALB/c mice (n = 5) was inoculated with heat-inactivated ZIKBeHMR-2, a second group (n = 15) received a single dose of 5 log PFU of ZIKBeHMR-2, and a third group (n = 14) received two doses of 5 log PFU of ZIKBeHMR-2 with a 6-week interval. All the serum specimens were heat-inactivated at 56 °C for 30 min. 

### 2.5. Plaque-Reduction Neutralization Test 

For the plaque-reduction neutralization test (PRNT), Vero cells were seeded in 24-well plates. Serum samples were 2-fold serially diluted in DMEM supplemented with 2% FBS with a starting dilution of 1:50, and incubated with an equal volume of a virus sample containing 100 PFU of MR766^GFP^ for 2 h at 37 °C. The virus-antibody mixture was added on cell monolayers for 2 h at 37 °C and then incubated with 0.8% CMC in growth medium. After a 4-day incubation, cell monolayers were fixed with 3.7% FA in PBS and then stained with 0.5% crystal violet in 20% ethanol. Visible plaques were manually counted. The neutralizing antibody titers were determined as the reciprocal of the last serum dilution that resulted in a 50% residual of infectivity of negative-control serum samples. PRNT_50_ values were determined from a nonlinear regression analysis.

### 2.6. Flow Cytometry Assay and Flow Cytometry-Based Neutralization Test 

For flow cytometry assay, cells were seeded in 24-well plates. Virus samples were added to the cells for 18 h. Cells were gently harvested by trypsinization, fixed with 3.7% FA in PBS, and permeabilized with 0.15% Triton X-100 in PBS for 5 min. Samples were incubated with mouse anti-flavivirus E MAb 4G2 at dilution 1:1000 (R&D Biotech, Besançon, France) and then with Alexa 647-conjugated anti-mouse IgG antibody (Abcam, Cambridge, UK) at dilution 1:1000. Cells were analyzed with a Cytoflex flow cytometer (Beckman Coulter, Villepinte, France). 

For flow cytometry-based neutralization test (FNT), 20,000 cells per well were seeded in 96-well plates. Serum specimens were 2-fold serially diluted in DMEM supplemented with 2% FBS with a starting dilution of 1:50, and incubated with an equal volume of a virus sample containing 20,000 PFU of MR766^GFP^ for 2 h at 37 °C. The virus-antibody mixture was added to the cell monolayers for 20 h. Cells were fixed with 3.7% FA in PBS, and 10,000 cells of each assay were analyzed for GFP expression with a CytoFLEX flow cytometer (Beckman Coulter, Villepinte, France). The 100% infectivity was obtained with the number of ZIKV-infected cells positive for GFP expression in the absence of serum samples. The neutralizing antibody titer was determined as the serum dilution that resulted in a 50% reduction of GFP expression (FNT_50_) and was calculated by nonlinear, dose-response regression analysis.

### 2.7. Statistical Analysis

Linear regression was used to determine the correlation, with associated *p*-value, between PRNT_50_ values and FNT_50_ values. Pearson’s correlation coefficients (r) and the coefficient of determination (R²) were determined using Prism software version 7.01 (GraphPad, San Diego, CA, USA).

## 3. Results 

### 3.1. Generation of Viral Clone MR766^GFP^ for Flow Cytometry Neutralization Test 

To generate a more reliable GFP reporter ZIKV for flow cytometry neutralization test, recombinant ZIKV_GFP_ was repeatedly passaged on Vero cells using a limiting dilution cloning method [13]. A single virus clone, hereafter named MR766^GFP^, was selected for further studies. The progeny virus production of MR766^GFP^ variant in Vero cells was at least 1.5 log higher than for parental ZIKV_GFP_ [13]. Expression of MR766^GFP^ was examined in Vero cells infected at a multiplicity of infection (m.o.i.) of 1 (Figure 1). We observed that most of the virus plaques produced in the focus-forming assay were positively stained by anti-GFP antibody (Figure 1A). Flow-cytometry analysis confirmed that Vero cells positive for GFP expression were recognized by the anti-E mAb 4G2 (Figure 1B). Thus, the MR766^GFP^ variant is a suitable GFP reporter ZIKV for the development of a FNT.

### 3.2. Comparison Between MR766^GFP^-Based FNT Assay and Conventional PRNT 

We reported that immunocompetent adult mice that received intraperitoneally 5 log PFU of chimeric ZIKV clone ZIKBeHMR-2 developed high titers of ZIKV-neutralizing antibodies [23]. ZIKBeHMR-2 is a chimeric clone of ZIKV derived from historical African ZIKV strain MR766 in which the structural protein region was replaced with that of epidemic ZIKV strain BeH819015 of Asian lineage [23]. Noteworthy, the ZIKBeHMR-2 E protein lacks N-glycosylation [23].

The neutralization assays were assessed with a selection of 29 immune sera from adult BALB/c mice that received one or two doses of ZIKBeHMR-2 with an interval of 6 weeks [23]. Immune sera from mice inoculated with heat-inactivated ZIKBeHMR-2 served as negative controls [23]. The serum samples were first subjected to testing by a conventional PRNT method in Vero cells using MR766^GFP^ (Figure 2). As expected, negative control immune sera exhibited no reduction of virus plaque number at the lower dilution tested (1:50). In contrast, immune sera from mice inoculated with ZIKBeHMR-2 showed neutralizing activity against MR766^GFP^. The serum samples reduced the number of virus plaques in a dilution-dependent manner. In mice that received a single dose of ZIKBeHMR-2, there was a PRNT_50_ value of 130 (95% Confident Interval (CI): 93–182). After inoculation of a booster dose, the ZIKV-neutralizing antibody titers increased up to 3051 (95% CI: 2251–4136). Thus, a conventional PRNT confirmed the neutralizing activity of anti-ZIKBeHMR-2 immune sera against ZIKV.

The mouse serum samples were next assessed by MR766^GFP^-based FNT. The neutralization tests were performed on Vero cells grown in 96-well plates, and GFP expression was examined 20 h post-infection (Figure 3). By flow-cytometry analysis, the percentage of GFP-positive Vero cells infected with MR766^GFP^ reached 55%. There was no reduction in the percentages of GFP-positive cells using immune sera from mice inoculated with heat-inactivated ZIKBeHMR-2 that served as a negative virus control (Figure 3A). Whereas mouse anti-ZIKBeHMR-2 immune sera showed neutralizing activity against MR766^GFP^ in a dilution-dependent manner (Figure 3B,C). In mice (n = 15) that received a single dose of ZIKBeHMR-2, there was a FNT_50_ value of 117 (95% CI: 81–169). After inoculation of a booster dose, the FNT_50_ values of immune sera (n = 14) increased up to 2469 (95% CI: 1673–3645).

Data fitting between PRNT_50_ and FNT_50_ values on anti-ZIKBeHMR-2 immune sera revealed a linear relationship (R² = 0.90, *p*-value <0.0001) between the two assays (Figure 4). This indicates a concordance between conventional PRNT and a FNT method based on a single infection round. Such results support the utilization of MR766^GFP^-based FNT for measuring ZIKV-neutralizing antibodies in serum specimens.

We next evaluated whether the MR766^GFP^-based FNT can be performed in human cells that do not support virus plaque assay such as human epithelial A549 cells. To evaluate the performance of our FNT on A549 cells, MR766^GFP^ was incubated with serial dilutions of immune sera from mice inoculated with one or two doses of ZIKBeHMR-2. The virus-antibody mixtures were used to infect A549 cells seeded in 96-well plates. At 20 h post-infection, the GFP expression was observed in infected A549 cells by flow cytometry analysis. The percentage of GFP-positive A549 cells infected with MR766^GFP^ reached 50%. There was a 100% infectivity with immune sera directed against heat-inactivated ZIKBeHMR-2 (Figure 5A). We observed that anti-ZIKBeHMR-2 immune sera reduced MR766^GFP^ infectivity for A549 cells in a serum dilution-dependent manner (Figure 5B,C). In mice (n = 15) that received a single dose of ZIKBeHMR-2, there was a FNT_50_ value of 161 (95% CI: 119–216). After inoculation of a booster dose, the FNT_50_ value of immune sera (n = 14) increased up to 2320 (95% CI: 1507–3570). 

As shown in Figure 6, there was a linear relationship (R² = 0.94, *p*-value <0.0001) between FNT_50_ performed in Vero and A549 cells. Thus, human epithelial A549 cells are suitable for the measurement of ZIKV-neutralizing antibodies in serum specimens based on our newly developed MR766^GFP^-based FNT.

## 4. Discussion

The development of new neutralizing assays that overcome the restrictions presented by the conventional PRNT methods become a priority for ZIKV. To this aim, we took advantage of our previously developed GFP reporter ZIKV (recombinant ZIKV_GFP_), which had been demonstrated to be a powerful tool for the monitoring of virus replication as well as the screening of antiviral compounds [13,14,15,16,17,18,19,20]. In routine, ZIKV_GFP_ infection is observed by flow cytometry analysis at 18–24 h post-infection based on the GFP signal. In the present study, we identified MR766^GFP^ as a variant of ZIKV_GFP_ exhibiting a greater propensity to produce a GFP signal associated with a higher virus progeny production.

Here, MR766^GFP^ was assayed with serum specimens from adult BALB/c mice inoculated with a chimeric viral clone ZIKBeHMR-2, which contains the structural protein region of epidemic Brazilian ZIKV strain BeH810915 into a MR766 backbone [23]. ZIKBeHMR-2 has a non-glycosylated E protein due to the introduction of a limited number of amino acid substitutions in the E glycan-loop of BeH819015 [23]. The ability of ZIKBeHMR-2 to induce anti-ZIKV antibody production had been validated in adult BALB/c mice immunized by the intraperitoneal route [23]. The serum specimens obtained from mice inoculated with one or two doses of ZIKBeHMR-2 were evaluated for neutralizing antibodies by conventional PRNT. Immunized mice developed anti-ZIKV antibodies that neutralize MR766 with PRNT_50_ values reaching up to 3500 [23]. Consequently, anti-ZIKBeHMR-2 immune sera were suitable for a comparative analysis between our MR766^GFP^-based FNT and conventional PRNT, the latter being considered as a “gold standard” assay for flavivirus neutralization. FNT was first performed on African green monkey Vero cells which are commonly used for neutralization tests. Vero cells were grown in 96-well-plates. The neutralizing activity of anti-ZIKBeHMR-2 immune sera against ZIKV was estimated within two days. There was a linear correlation between FNT_50_ and PRNT_50_ values with R^2^ coefficient of determination of 0.9. Thus, our newly developed GFP reporter ZIKV-based FNT allows a prompt and reliable detection of ZIKV-neutralizing antibodies in serum specimens. A such method was next employed on human epithelial A549 cells, and there was a linear correlation between FNT_50s_ values in A549 and Vero cells with an R^2^ coefficient of determination of 0.94. We propose that MR766^GFP^-based FNT is a suitable method for the general measurement of antibody-mediated ZIKV neutralization. Now, it is of paramount importance to estimate our newly developed FNT format for the detection of ZIKV-neutralizing antibodies using human serum specimens from Zika patients.

## 5. Conclusions

In the present study, we demonstrated that viral clone MR766^GFP^ is a suitable GFP reporter ZIKV for the identification of ZIKV-neutralizing antibodies in serum specimens by flow-cytometry analysis. The time-saver MR766^GFP^-based FNT format could provide a reliable platform for measuring antibodies that neutralize ZIKV in a large number of serum specimens without the requirement for plaque-forming formation. Further studies will be necessary to determine whether conventional PRNT method might be replaced by our newly developed MR766^GFP^-based FNT for epidemiological cohort studies as well as vaccine development on ZIKV.

## 6. Patent

The viral clone ZIKBeHMR-2 has been described in the patent entitled “Vaccine compositions comprising an attenuated mutant Zika virus” under the number WO2017220748A1 (priority date 2016-06-23).

## Figures and Tables

**Figure 1 vaccines-07-00066-f001:**
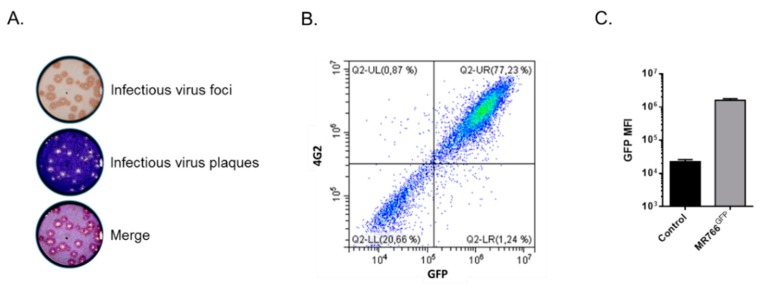
Characterization of viral clone MR766^GFP^ on Vero cells. In (**A**), the example of infectious virus foci developed after a focus-forming assay using anti-GFP antibody. To visualize infectious virus plaques, the cell monolayer was counterstained with crystal violet. In (**B**), cells were infected 24 h with MR766^GFP^ at an m.o.i. of 1. The GFP-positive cells (GFP) stained with anti-E MAb 4G2 (4G2) were analyzed by flow cytometry analysis. In (**C**), mean fluorescent intensity (MFI) of MR766^GFP^-infected cells (MR766^GFP^), as determined by flow cytometry analysis. Control: mock-infected Vero cells.

**Figure 2 vaccines-07-00066-f002:**
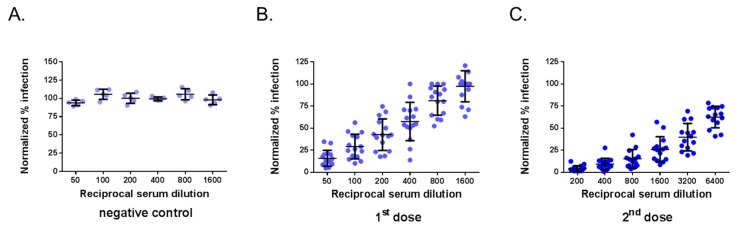
Plaque-reduction neutralization test on anti-Zika virus immune sera. The neutralizing ability of mouse serum specimens against MR766^GFP^ was determined by plaque-reduction neutralization test (PRNT) on Vero cells. Samples of anti-ZIKBeHMR-2 immune sera were tested individually. Serum specimens were two-fold serial diluted starting at a 1:50 dilution. The results are shown normalized to the 100% infection achieved with MR766^GFP^ without serum. In (**A**), the immune sera of BALB/c mice inoculated with heat-inactivated ZIKBeHMR-2 served as a negative control (negative control). In (**B**), immune sera of BALB/c mice that received a single dose of ZIKBeHMR-2 (5 log PFU) (1^st^ dose). In (**C**), immune sera of BALB/c mice that received two doses of ZIKBeHMR-2 (5 log PFU) with an interval of 6 weeks (2^nd^ dose).

**Figure 3 vaccines-07-00066-f003:**
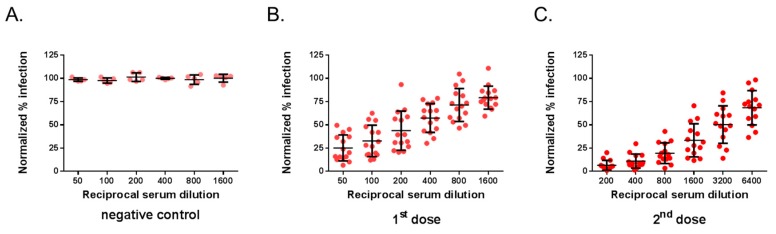
Flow cytometry neutralization test on Vero cells. The neutralizing ability of mouse serum specimens against ZIKV was determined by flow cytometry neutralization test (FNT) on Vero cells. Samples of anti-ZIKBeHMR-2 immune sera were tested individually. Serum specimens were two-fold serial diluted starting at a 1:50 dilution and mixed with an equal volume of virus sample containing 20,000 PFU of MR766^GFP^. The virus-antibody mixture was added on 20,000 Vero cells grown in 96-well plates. At 20 h post-infection, the percentage of positive cells for GFP expression was determined by flow-cytometry analysis. The number of GFP-positive cells without serum was considered 100% of infectivity. The results were expressed as the percentage of GFP-positive cells in assay relative to that calculated in absence of serum. In (**A**), the immune sera of BALB/c mice inoculated with heat-inactivated ZIKBeHMR-2 served as a negative control (negative control). In (**B**), immune sera of BALB/c mice that received a single dose of ZIKBeHMR-2 (5 log PFU) (1^st^ dose). In (**C**), immune sera of BALB/c mice that received two doses of ZIKBeHMR-2 (5 log PFU) with a 6-week interval (2^nd^ dose).

**Figure 4 vaccines-07-00066-f004:**
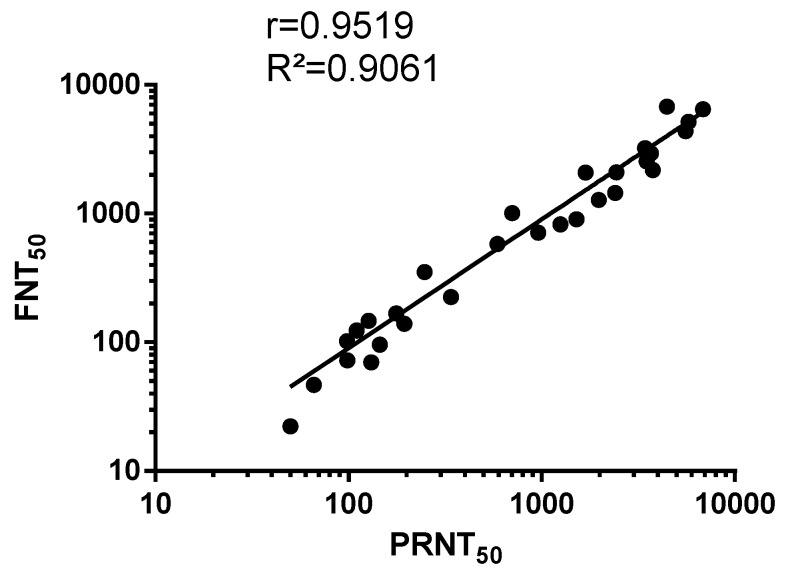
Scatter plots of PRNT_50_ and FNT_50_ values in Vero cells. PRNT: plaque-reduction neutralization test; FNT: flow-cytometry neutralization test. The PRNT_50_ and FNT_50_ were determined as the reciprocal of the last serum dilution that resulted in a 50% residual of infectivity of negative-control serum samples.

**Figure 5 vaccines-07-00066-f005:**
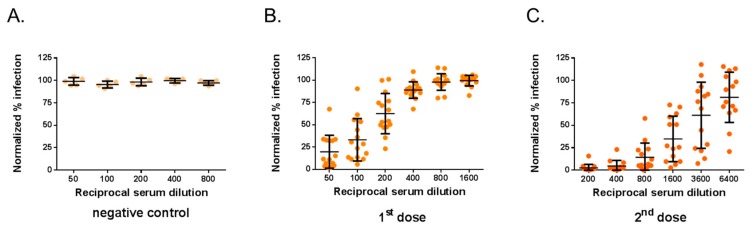
Flow cytometry neutralization test on A549 cells. The neutralizing ability of mouse serum specimens against ZIKV was determined by FNT on A549 cells. Samples of anti-ZIKBeHMR-2 immune sera were tested individually. Serum samples were two-fold serial diluted starting at a 1:50 dilution and mixed with an equal volume of virus sample containing 20,000 PFU of MR766^GFP^. The virus-antibody mixture was added on 20,000 A549 cells grown in 96-well plates. At 20 h post-infection, the percentage of positive cells for GFP expression was determined by flow-cytometry analysis. The number of GFP-positive cells without serum was considered 100% of infectivity. The results were expressed as the percentage of GFP-positive cells in assay relative to that calculated without serum. In (**A**), The immune sera of BALB/c mice inoculated with heat-inactivated ZIKBeHMR-2 served as a negative control (negative control). In (**B**), immune sera of BALB/c mice that received a single dose of ZIKBeHMR-2 (5 log PFU) (1^st^ dose). In (**C**), immune sera of BALB/c mice that received two doses of ZIKBeHMR-2 (5 log PFU) with a 6-week interval (2^nd^ dose).

**Figure 6 vaccines-07-00066-f006:**
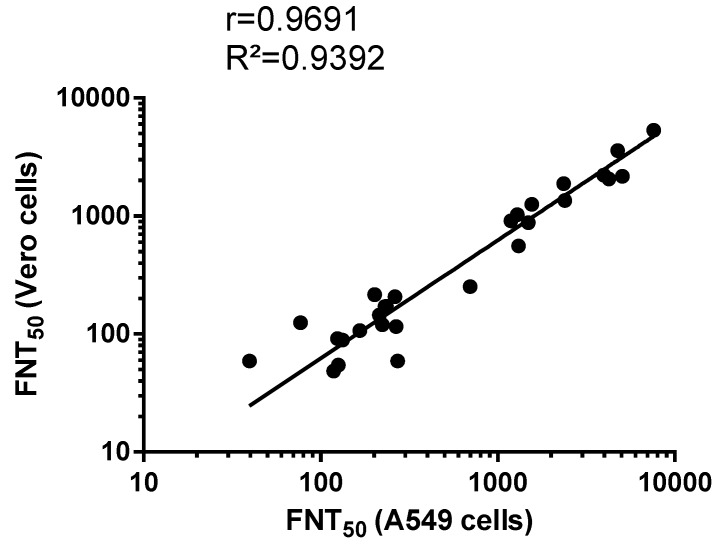
Scatter plots of FNT_50_ on Vero and A549 cells. FNT: flow-cytometry neutralization test. The FNT_50_ were determined as the reciprocal of the last serum dilution that resulted in a 50% residual of infectivity of negative-control serum samples.

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
