# Peer review of "A GFP Reporter MR766-Based Flow Cytometry Neutralization Test for Rapid Detection of Zika Virus-Neutralizing Antibodies in Serum Specimens"

_vaccines, 2019, doi:10.3390/vaccines7030066_

Round 1
Reviewer 1 Report
This research article by Frumence et al focuses on GFP reporter MR766-based flow-cytometry 2 neutralization test (FNT) as alternative to the time consuming and labor intensive PRNT assay. Data fitting between PRNT50 and FNT50 values on anti-ZIKBeHMR-2 immune sera revealed a linear relationship (figure 4) which demonstrated the efficiency and convenience of this assay. The method is valid and substantial. Only thing needed to think of is if it is applicable to assess the breadth of serum sample.
Author Response
July 12nd 2019
Ref: vaccines-546525
Dear Editor,
Thank you for your high consideration of our submitted manuscript # vaccines-525906.
Three minor changes made to the revised manuscript can be found below.
#1. We have stated on the opportunity or not to test our FNT method for the breadth of serum sample.
#2. We provide data on the multiplicities of infection used in our study and the percentages of ZIKV-infected cells positive for GFP expression determined by flow-cytometry analysis.
#3. We have included more references on the advantages using our GFP reporter ZIKV for screening of antiviral molecules.
We greatly appreciate the efforts of the reviewers and the editorial team of Vaccines to help us make this the strongest work, and we look forward to your reply.
Thank you very much for your assistance and we look forward to your earliest reply.
Sincerely yours,
Philippe DESPRES, PhD
La Reunion island University
& UM 134 PIMIT
CYROI, 97491 Sainte-Clotilde, France
philippe.despres@univ-reunion.fr
Responses to Reviewers 1 & 2:
We thank the two Reviewers for their thoughtful remarks and the opportunity to revise our manuscript. We believe that we have been able to address their minor concerns and that this has improved the manuscript. Please find below a list of the Reviewer’s comments, each of which is followed by our response.
# Reviewer 1:
This research article by Frumence et al focuses on GFP reporter MR766-based flow-cytometry neutralization test (FNT) as alternative to the time consuming and labor intensive PRNT assay. Data fitting between PRNT50 and FNT50 values on anti-ZIKBeHMR-2 immune sera revealed a linear relationship (figure 4) which demonstrated the efficiency and convenience of this assay. The method is valid and substantial. Only thing needed to think of is if it is applicable to assess the breadth of serum sample.
We appreciated this point raised by the reviewer #1. It is widely admitted that ZIKV-neutralizing antibodies are mainly directed against the envelope E protein carrying the neutralizing antibody epitopes at the virus surface. To date, it was not attempted that our FNT could be a suitable method for assessing the breadth of serum sample against various ZIKV antigens.
Reviewer 2 Report
Frumence E et al described the application of GFP reporter for virulent strain of Zika virus for FNT assay as comparison with gold-standard PRNT.
Overall manuscript is well written and clear. Details of methods are provided, well-established. This manuscript will be interested to the antiviral research and vaccine field.
Only minor points that perhaps can strengthen the manuscript.
The authors demonstrated in fig. 1 that moi of 1 was used to validate. Did this moi is used in the assays in fig. 2, 3, 4?
In related in Q1, if not , the authors should describe how many percentages of infected cells without sera, particular for fig.4 (A459 cells)
Author Response
July 12nd 2019
Ref: vaccines-546525
Dear Editor,
Thank you for your high consideration of our submitted manuscript # vaccines-525906.
Three minor changes made to the revised manuscript can be found below.
#1. We have stated on the opportunity or not to test our FNT method for the breadth of serum sample.
#2. We provide data on the multiplicities of infection used in our study and the percentages of ZIKV-infected cells positive for GFP expression determined by flow-cytometry analysis.
#3. We have included more references on the advantages for using our GFP reporter ZIKV for screening of antiviral molecules.
We greatly appreciate the efforts of the reviewers and the editorial team of Vaccines to help us make this the strongest work, and we look forward to your reply.
Thank you very much for your assistance and we look forward to your earliest reply.
Sincerely yours,
Philippe DESPRES, PhD
La Reunion island University
& UM 134 PIMIT
CYROI, 97491 Sainte-Clotilde, France
philippe.despres@univ-reunion.fr
Responses to Reviewers 1 & 2:
We thank the two Reviewers for their thoughtful remarks and the opportunity to revise our manuscript. We believe that we have been able to address their minor concerns and that this has improved the manuscript. Please find below a list of the Reviewer’s comments, each of which is followed by our response.
# Reviewer 2:
Frumence E et al described the application of GFP reporter for virulent strain of Zika virus for FNT. Overall manuscript is well written and clear. Details of methods are provided, well-established. This manuscript will be interested to the antiviral research and vaccine field.
Only minor points that perhaps can strengthen the manuscript.
The authors demonstrated in fig. 1 that moi of 1 was used to validate. Did this moi is used in the assays in fig. 2, 3, 4?
In related in Q1, if not, the authors should describe how many percentages of infected cells without sera, particular for fig.4 (A459 cells)
We appreciated the two points raised by the reviewer #2.
In Figure 2, for a conventional PRNT method, a virus sample containing 100 PFU of MR766GFP virus is mixed with an equal volume of serum sample and the remaining infectivity is then titrated on Vero cells. Consequently, we performed PRNT with a m.o.i of 0.001 and such assay relates to an infection of about 7.5 x 104 cells grown in 24-well plates with 100 PFU of ZIKV. For Figures 3 and 5, Vero or A549 cells were infected 20 h with MR766GFP virus at m.o.i. of 1. The figure legends were modified in the revised manuscript accordingly.
For Figures 3 and 5, there are about 55 % of GFP-positive cells among Vero cells infected 20 h with MR766GFP at moi of 1. For A549 cells infected with MR766GFP at moi of 1, 50% of GFP-positive cells were observed at 20h post-infection. The text was modified in the revised manuscript accordingly.